# Global ocean redox changes before and during the Toarcian Oceanic Anoxic Event

Alexandra Kunert [1] ✉ & Brian Kendall [1]

Mesozoic oceanic anoxic events are recognized as widespread deposits of marine organic-rich mudrocks temporally associated with mass extinctions and large igneous province emplacement. The Toarcian Oceanic Anoxic Event is one example during which expanded ocean anoxia is hypothesized in response to environmental perturbations associated with emplacement of the Karoo−Ferrar igneous province. However, the global extent of total seafloor anoxia and the relative extent of euxinic (anoxic and sulfide-rich) and non-euxinic anoxic conditions during the Toarcian Oceanic Anoxic Event are poorly constrained. Here we present estimates of the global total anoxic and euxinic seafloor areas before and during the Toarcian Oceanic Anoxic Event based on rhenium and molybdenum enrichments, respectively, in organic-rich mudrocks of the Fernie Formation (British Columbia, Canada). We find that mass balance models depict an expansion of up to ~7% total seafloor anoxia, which was dominated by euxinia, at the onset of the Toarcian Oceanic Anoxic Event, followed by a contraction before the end of the event. The global ocean redox trends revealed by the rhenium data mirrors the collapse and recovery patterns of global ammonite and foraminiferal biodiversity.

The ca. 183 Ma Toarcian Oceanic Anoxic Event (T-OAE) represents a significant perturbation to Earth's atmosphere−ocean system and was associated with a global mass extinction[1,2]. The event, lasting an estimated 300 kyr to >1 Myr[3–6], coincides with and is thought to have been triggered by emplacement of the subaerial Karoo−Ferrar large igneous province (LIP) in southern Pangea[1,3,7]. The environmental perturbances triggered by LIP emplacement included a dynamic climate[8–10], enhanced hydrological cycle and continental weathering[10–13], increased organic marine productivity[3], and a significant expansion of seafloor anoxia and euxinia[14,15]. Despite the OAE naming convention which emphasizes the importance of expanded seafloor anoxia, the global extent of total bottom water anoxia ($[O_2]_{aq} = 0$ ml l$^{-1}$)[16,17] across the T-OAE remains poorly constrained, as is the relative extent of non-euxinic anoxia (including ferruginous bottom waters where $Fe^{2+}$ accumulates[17–19]) and euxinia (anoxic, $H_2S$-rich bottom waters[17–19]).

The global extent of ocean euxinia during the T-OAE has been inferred from molybdenum isotope compositions ($\delta^{98}Mo$) in organic-rich mudrock (ORM). When the oceans are primarily oxic (abundant $O_2$ in bottom waters and penetrating >1 cm below the sediment–water interface), manganese (Mn) oxides enriched in lighter-mass Mo isotopes are buried over a larger portion of the seafloor, resulting in global seawater with a higher $\delta^{98}Mo$ as observed today. By contrast, the low $\delta^{98}Mo$ of locally euxinic ORM from multiple sections in Europe suggest that global seafloor euxinia expanded and oxic seafloor contracted during the early stages of the T-OAE[20–22]. Euxinic waters have been estimated to cover 2–10% of the seafloor during the T-OAE, which is one to two orders of magnitude greater than the modern ocean (~0.1–0.2%)[19,20,22]. However, suggestions that the Toarcian sections of Europe were deposited in sedimentary basins highly restricted from global ocean circulation have called these interpretations into question[9,23–25]. Importantly, Mo isotope data do not reliably constrain the extent of non-euxinic anoxia because the magnitude of isotope fractionation is similar during Mo removal to sediments under weakly euxinic, non-euxinic anoxic, and suboxidizing to dysoxic (low [<15 μM] to mild [60–90 μM] $O_2$ in bottom waters commonly penetrating <1 cm below the sediment–water interface[16,17]) settings[26].

Thallium isotope compositions ($\epsilon^{205}Tl$) from ORM have also been applied to reconstruct the global ocean redox landscape during the

[1]Department of Earth and Environmental Sciences, University of Waterloo, Waterloo N2L 3G1, Canada. ✉e-mail: akunert@uwaterloo.ca

T-OAE. Preferential burial of isotopically heavy Tl adsorbed to Mn oxides deposited in oxic seafloor sediments shifts seawater towards lower (more negative) $\varepsilon^{205}$Tl, whereas increased anoxic seafloor area will shift seawater to higher (less negative) $\varepsilon^{205}$Tl. In the early Toarcian, two distinctive shifts towards higher $\varepsilon^{205}$Tl are captured in ORM from western Canada, one ~500 kyr before a negative carbon isotope excursion (N-CIE) and another at the onset of the N-CIE, which are thought to indicate two expansions of ocean anoxia[27]. The highest $\varepsilon^{205}$Tl values at the onset of the N-CIE indicate that burial of Mn oxides decreased by at least 50% during the T-OAE, suggesting an equivalent increase of dysoxic, suboxidizing, anoxic but non-euxinic, and/or euxinic bottom waters[27]. However, the Tl isotope proxy does not distinguish the relative seafloor areas covered by these $O_2$-deficient redox settings.

An alternative approach to quantify the overall extent of oceanic anoxia incorporating the distinction between non-euxinic anoxic and euxinic seafloor during the T-OAE is required. Previously, redox-sensitive elemental mass balances[18,19,28] have been used to determine the extent of total anoxic (uranium, U; chromium; rhenium, Re) or euxinic (Mo) seafloor in the Proterozoic Eon. However, no known studies have attempted to calibrate and apply these models across shorter timescales such as the T-OAE, although drawdown of the global dissolved reservoir of redox-sensitive trace metals during the T-OAE has been inferred[20,29,30]. Here, we use temporally calibrated Re[18] and Mo[19] elemental mass balances together to quantitively estimate both total anoxic seafloor area (non-euxinic + euxinic) and euxinic seafloor area, respectively, for Pliensbachian to Toarcian ORM (Gordondale Member, Fernie Formation, British Columbia, Canada; Fig. 1). We also take advantage of the lower oceanic residence time of Re to look for evidence of contraction in the extent of anoxic seafloor prior to the end of the T-OAE. Previous work using Mo and sulfur (S) isotopes has revealed that contraction of euxinic seafloor occurred after the T-OAE[20–22,31], but the longer oceanic residence times of these elements means a slower response time to global redox changes. Thallium has an even shorter residence time than Re making sedimentary Tl isotope data potentially vulnerable to influence by basin restriction[32]. It is therefore possible that interpretation of global redox variation by Tl isotopes in some sections may be skewed by local depositional conditions.

Both Re and Mo are ideal trace metals to quantify global redox conditions in the oceans due to their conservative behavior in oxygenated seawater, long ocean residence time (130 kyr for Re and 440 kyr for Mo in modern seawater)[33], and low detrital background in organic-rich marine sediments[18,19]. High Re burial rates, and thus Re

enrichments, in marine sediments occur in both non-euxinic anoxic and euxinic environments via complexation of Re with organic matter and sulfide minerals[34], but negligible Re enrichment occurs in oxic sediments. Molybdenum preferentially forms thiomolybdate complexes in the presence of dissolved $H_2S$[35] and the burial efficiency of sulfurized Mo species in euxinic sediments is significantly elevated compared to Mo burial in non-euxinic settings[19]. Notably, modern continental margin sediments with mild/low $O_2$ concentrations in bottom waters and <1 cm $O_2$ penetration into sediments are known to have elevated Re enrichments but minimal Mo enrichments, highlighting that Re is not dependent solely on dissolved sulfide for burial in sediments[36–39]. This observation is key to the premise that the sedimentary Re record provides insight on the total anoxic seafloor area globally whereas Mo constrains the euxinic seafloor area. These distinctive redox characteristics mean that a predominantly oxic global ocean will have large dissolved Re and Mo reservoirs sourced from rivers, and thus pronounced enrichments will occur in ORM that cover small areas of the open ocean within oxygen minimum zones on continental margins. By contrast, an expansion of anoxic and euxinic marine sediment sinks will draw down the dissolved oceanic Re and Mo reservoirs, respectively, leading to muted enrichments in ORM deposited in unrestricted marine settings[18,19]. Using these principles, Re and Mo oceanic mass balance models can be used to infer the extent of seafloor total anoxia and euxinia, respectively, using their enrichments in locally anoxic or euxinic ORM deposited in unrestricted marine settings. The extent of non-euxinic anoxic seafloor can be determined as the difference between the total anoxic and euxinic seafloor areas.

## Results and discussion

### Local paleoenvironmental setting of the T-OAE in British Columbia

The T-OAE is expressed globally by synchronous deposits of ORM recording a N-CIE in organic and carbonate carbon, typically embedded within a broader positive CIE[14,15,40–46]. The T-OAE was identified in the Red Deer Member (Fernie Formation) of Alberta, Canada in cores (1-35-62-5W6 and 6-32-75-20W5) and outcrop (Bighorn Creek East Tributary) by a N-CIE with a magnitude of −3‰ to −4‰[40,47]. We report new elemental and organic carbon isotope data ($\delta^{13}C_{org}$) from drill core c-B6-A/94-B-8 in British Columbia, Canada (provided in the Supplementary Dataset 1), that hosts equivalent Fernie Formation strata (Gordondale Member). This core contains a N-CIE across a vertical interval of 7.1 m and has a magnitude of −2.0‰ to −2.4‰ (Fig. 2a). Using this $\delta^{13}C_{org}$ profile, the section can be subdivided into informal

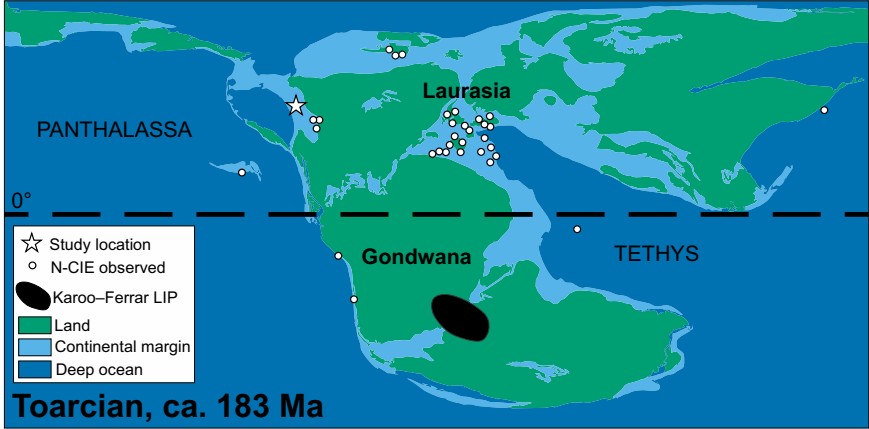

**Fig. 1 | Global Toarcian paleogeography.** Negative carbon isotope excursion locations[40] used to identify the Toarcian Oceanic Anoxic Event are shown. The core used in this study was deposited along the western margin of Laurasia (proto-North America) and was connected to the Panthalassa ocean. Paleogeography modified from the PaleoMAP Project (Scotese, 2021)[82].

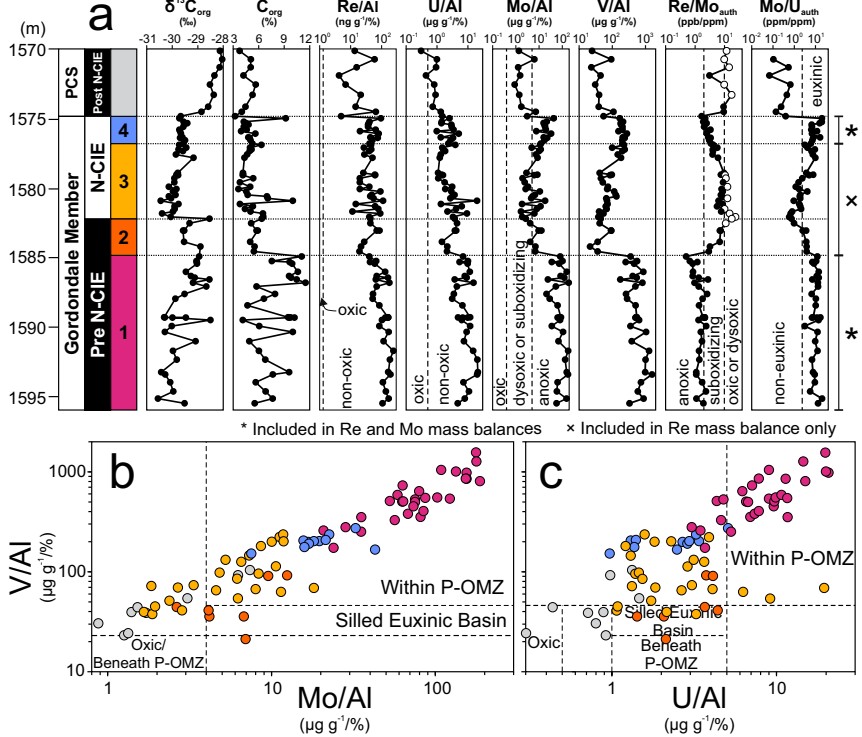

**Fig. 2 | Geochemical data for the c-B6-A/94-B-8 core hosting T-OAE deposits.** Gordondale Member core (**a**) geochemical profiles, (**b**) V/Al versus Mo/Al covariation, and (**c**) V/Al versus U/Al covariation. The Toarcian Oceanic Anoxic Event is classically recognized by a negative carbon isotope excursion (N-CIE), which is observed in the organic carbon fraction ($\delta^{13}C_{org}$) between 1575.00 and 1582.10 m depth. Trace metal/Al, and authigenic (auth) Re/Mo and Mo/U ratios are used to characterize the local depositional redox conditions and select suitable samples for global Re and Mo mass balance models. Thresholds for each proxy are shown as dashed vertical lines on the profiles and covariations as discussed in the text. Environments within or beneath perennial (P-) continental margin oxygen minimum zones (OMZs), normal oxic settings and silled euxinic basins are noted on the covariation diagrams. Thresholds and covariation fields after Bennett and Canfield[36]. White circles on the $Re/Mo_{auth}$ profile are interpreted as oxic/dysoxic and are not included in the global mass balance models. The Poker Chip Shale (PCS) is not included in the models due to overwhelmingly oxic/dysoxic deposition. Model intervals (MI) discussed in the text are labeled from MI-1 to MI-4 upsection.

intervals relative to the N-CIE: Pre-N-CIE (>1582.10 m) and N-CIE (1582.10–1575.00 m) within the Gordondale Member and Post-N-CIE (<1575.00 m) of the overlying Poker Chip Shale (PCS). A second N-CIE is observed below 1587.00 m but may represent an older Sinemurian or Pliensbachian event as observed in other sections in the basin[47,48]. Direct age constraints are not available within the study core, however, several sections of Early Jurassic age in Alberta between ~200 and 650 km from the studied section have biostratigraphic (ammonite) and geochronologic (Re–osmium [Os]; U–lead [Pb]) ages confirming the presence of Toarcian strata[11,40,49,50]. The $\delta^{13}C_{org}$ and gamma-ray correlations with those sections suggest that the studied drill core likely contains T-OAE deposits (see Fig. S1 in the Supplementary Information).

To evaluate global seafloor redox conditions across the T-OAE, local depositional bottom water redox conditions must be established. The global seawater mass balances function with the assumption that, in non-restricted settings, Re enrichments in sediments accumulating under generally anoxic (non-euxinic or euxinic) local bottom water conditions, and Mo enrichments in sediments accumulating under locally euxinic bottom waters, reflect the size of the global aqueous seawater reservoirs of these trace metals[18,19]. Thus, identifying local redox conditions where these trace metals best reflect the global seawater reservoir is essential.

Local bottom water redox conditions during deposition of the Gordondale Member in our study section can be constrained using well-established redox-sensitive trace metal proxies. These metals, which include the targeted metals in our study (Re, Mo) and others like U and vanadium (V), are soluble in oxygenated seawater and have low burial rates into oxygenated (including oxic, dysoxic and occasionally suboxidizing) sediments with low $C_{org}$ accumulation, but larger burial rates in low $O_2$ environments with elevated $C_{org}$[36,51]. Both Mo and V require bottom water euxinia for large authigenic enrichments ($X_{auth}$; Equation S1) in sediments, whereas Re and U do not[28,36,37,52–57]. Thus, low or mild $V_{auth}$ and $Mo_{auth}$ but elevated $U_{auth}$ and $Re_{auth}$ in sediments suggest deposition from locally non-euxinic anoxic or suboxidizing bottom waters[28,37]. Elevated enrichment of all four metals[54,56,57] suggests deposition from locally euxinic bottom waters[54,56,57].

Due to the contrasting enrichment styles of Mo and V with U and Re, and possible variation in absolute metal concentrations depending on local environmental factors (e.g., sedimentation rate, organic carbon content), ratios of these metals against one another, or normalized to aluminum (Al) provide the most robust method for paleoredox interpretations[36,37,58]. We take advantage of a recent study that compiled trace metal data from modern anoxic basins (oxic margins, oxygen minimum zones [OMZs] and silled euxinic basins)[36] to place estimates on redox thresholds. Oxic environments include those from continental margins (herein referred to as "normal oxic"), or beneath seasonal OMZs (S-OMZ)—all have $O_2 > 105\,\mu M$. Suboxidizing environments are found within the perennial OMZ (P-OMZ) of the Mexican margin and part of the Peruvian margin where $O_2$ is <5 μM[36]. Fully anoxic environments include areas within the P-OMZ of the Peruvian and Namibian margins, and the silled euxinic basins of the Black Sea and Cariaco Basin. Areas below P-OMZs feature mainly dysoxic bottom waters (all $O_2$ from 10 to 100 μM; dysoxic is defined as 15 to 60–90 μM) and thus are treated as representative of dysoxic settings. Areas within S-OMZ are not included here as $O_2$ was reported as seasonally variable. The $Re/Mo_{auth}$ and $Mo/U_{auth}$ ratios were not directly reported in Bennett and Canfield[36], however, these ratios have been used elsewhere

for local paleoredox interpretations[37,59–61]. Thus, we calculated these ratios from the Bennett and Canfield data and determined thresholds for each redox setting from the compiled data using a similar statistical method as done by Bennett and Canfield[36], minimizing false positives.

Both normal oxic environments and oxic areas below S-OMZs are characterized by low trace metal/Al ratios, most distinctly by Re/Al (<1.3 ppb/%), Mo/Al (<0.4 ppm/%) and U/Al (<0.5 ppm/%)[36]. However, normal oxic environments feature highly variable Re/$Mo_{auth}$ (1.2 to 31 ppb/ppm) and Mo/$U_{auth}$ (0.02 to 26 ppm/ppm), while oxic areas beneath S-OMZs record persistently high Re/$Mo_{auth}$ (>26 ppb/ppm) and low Mo/$U_{auth}$ (<0.3 ppm/ppm). Dysoxic sediments below P-OMZs have similar Re/$Mo_{auth}$ and Mo/$U_{auth}$ to the areas under S-OMZs (>21 ppb/ppm and <0.3 ppm/ppm, respectively), but are distinguished from all other settings by V/Al (<23 ppm/%) and U/Al (1–5 ppm/%)[36]. Suboxidizing environments within some P-OMZs where $O_2$ < 5 μM are not easily distinguished by Al-normalized trace metal data, however, we observe that Re/$Mo_{auth}$ ratios of 2–10 ppb/ppm can be used to characterize this setting. We select a conservative threshold of Re/$Mo_{auth}$ > 10 ppb/ppm to distinguish the oxic and dysoxic areas below S- and P-OMZs from the suboxidizing areas within P-OMZs (<10 ppb/ppm) as only a single sample from a suboxidizing setting lies between 10–20 ppb/ppm (10.1 ppb/ppm). Anoxic waters within P-OMZs and silled euxinic basins both feature elevated Mo/Al and lower Re/Mo compared to the more oxygenated environments (>5 ppm/% and <2 ppb/ppm, respectively)[36]. These environments can be differentiated using V/Al (P-OMZ > 46 ppm/%; silled euxinic basin = 23–46 ppm/%) and U/Al (P-OMZ > 5 ppm/%; silled euxinic basin = 2–5 ppm/%)[36], as well as Mo/$U_{auth}$ when an oxygenated environment is ruled out (P-OMZ < 2.4 ppm/ppm < silled euxinic basin). The Re/$Mo_{auth}$ thresholds determined here for the oxic/dysoxic–suboxidizing (10 ppb/ppm) and suboxidizing–anoxic (2 ppb/ppm) thresholds align closely with those of previous studies (~15 ppb/ppm and ~4 ppb/ppm, respectively[59]).

The goal of the local redox classification is to select viable samples for the global Re and Mo mass balance models. The greatest authigenic Re enrichments occur in sediments here classified as suboxidizing (within P-OMZs with $O_2$ < 5 μM), and in fully anoxic environments. The highest authigenic Mo enrichments only occur in fully anoxic environments, with considerable overlap in Mo/Al ratios between the P-OMZ and silled euxinic basins. Molybdenum is typically depicted as an indicator of euxinia, however, bottom water $H_2S$ levels were not reported for the compiled dataset in Bennett and Canfield[36]. Euxinia has been reported within the Namibian P-OMZ (ref. [62]), and although this has not been reported at sites studied within the Peruvian P-OMZ (e.g., ref. [63]), it is probable that areas within ancient P-OMZs experienced at least intermittent euxinia. Thus, we select only those intervals that are classified as suboxidizing or anoxic for the Re model, or fully anoxic with euxinia for the Mo model.

The Gordondale Member (Pre-N-CIE and N-CIE intervals) hosts elevated trace metal/Al ratios and $C_{org}$ (Fig. 2a), suggesting generally low $O_2$ conditions during deposition. However, variability in the degree of enrichment exists across the section. Oxic samples are rare, only occurring periodically in the overlying PCS (Post-N-CIE interval) which records generally oxic to dysoxic conditions based on low U/Al and high Re/$Mo_{auth}$ and plotting towards the oxic/dysoxic fields (lower left) of the V/Al versus Mo/Al (Fig. 2b) and U/Al (Fig. 2c) covariations. Similar dysoxic conditions are observed as elevated Re/$Mo_{auth}$ at the boundary between the Pre-N-CIE and N-CIE intervals and within the lowermost N-CIE. All other samples were likely deposited under at least suboxidizing conditions. The lowermost Pre-N-CIE and uppermost N-CIE intervals both feature elevated Mo/Al and low Re/$Mo_{auth}$ and plot well into the anoxic field (upper right) of the covariations involving Mo/Al, V/Al, and U/Al (Fig. 2b, c) such that these intervals were likely deposited under fully anoxic conditions.

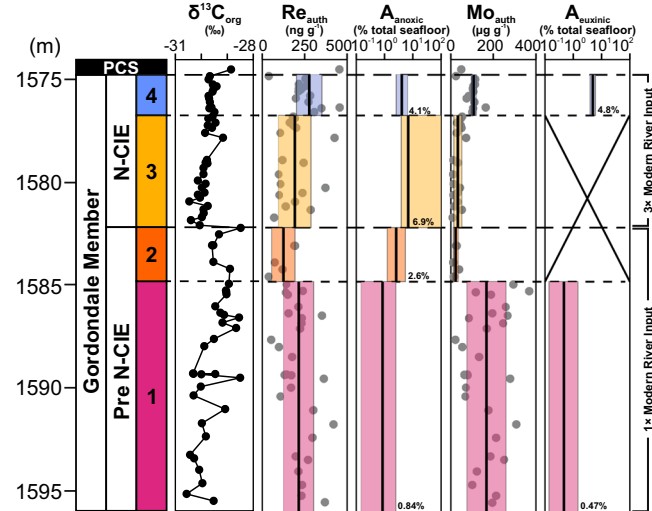

**Fig. 3 | Rhenium and molybdenum mass balance model results based on Gordondale Member authigenic metal enrichments to determine seafloor areas of anoxia ($A_{anoxic}$) and euxinia ($A_{euxinic}$) prior to and during the Toarcian Oceanic Anoxic Event (T-OAE).** All model results were determined using a bulk mass accumulation rate calculated from the median duration of the T-OAE (400 kyr)[3–5]. Pre-N-CIE model results (model intervals, MI-1 and MI-2) were determined with river inputs set to modern ($4.29 \times 10^5$ mol Re yr$^{-1}$; $3.00 \times 10^8$ mol Mo yr$^{-1}$)[18,19,33], while N-CIE intervals (MI-3 and MI-4) were determined with river input set to 3-times modern[11]. No results for the Mo mass balance modeling are presented for MI-2 or MI-3 because the minimal enrichments indicate local redox conditions (non-euxinic anoxic or suboxidizing) that do not reflect changes to the global dissolved seawater Mo reservoir. Mean values are shown as solid black lines, bracketed by 1 standard deviation shaded boxes. Abbreviations: m, meters; PCS, Poker Chip Shale; N-CIE, negative carbon isotope excursion.

The Mo/$U_{auth}$ of these anoxic samples is typically >2.4 ppm/ppm, suggesting that there was $H_2S$ present in the bottom waters of a P-OMZ. It is not likely our samples were deposited in a silled euxinic basin because most samples do not plot in the silled euxinic basin fields defined by Mo/Al, V/Al, and U/Al ratios. The intervening samples (uppermost Pre-N-CIE and lowermost N-CIE) do not share agreement between each redox indicator, e.g., Re/$Mo_{auth}$ indicates mainly suboxidizing, Mo/Al suggests some anoxic deposition, and U/Al suggests that samples were not deposited under oxic conditions. Therefore, this interval was likely deposited under suboxidizing to anoxic non-euxinic bottom waters.

Based on the geochemical proxies, the study section, which was initially divided into Pre-N-CIE, N-CIE, and Post-N-CIE from the $\delta^{13}C_{org}$ profile, can be further subdivided by local redox changes to identify intervals most suitable for application of the Re and Mo global mass balances and to allow an opportunity to examine global ocean redox changes at finer temporal resolution. Both Pre-N-CIE and N-CIE intervals include suboxidizing/non-euxinic anoxic conditions suitable for the Re mass balance model, and euxinic conditions suitable for both Re and Mo models. Hence, each is divided into lower and upper portions according to redox state. The lower Pre-N-CIE, hence referred to as Model Interval 1 (MI-1) is interpreted as dominantly euxinic, the upper Pre-N-CIE (MI-2) and lower N-CIE (MI-3) as suboxidizing or anoxic but non-euxinic, and the upper N-CIE (MI-4) as euxinic. Because of the inexact agreement between the redox proxies discussed above, these interval boundaries were selected based on the Mo/Al (Fig. 2a) and $Mo_{auth}$ (Fig. 3) data to avoid biasing the euxinic intervals with samples of improbably low Mo enrichments that would lead to overestimates of the global seafloor area covered by euxinic bottom waters. No further geochemical subdivision is applied to the Post-N-CIE as most samples are dysoxic.

In addition to the V/Al versus Mo/Al and U/Al covariations, cross plots of Mo versus U and cadmium (Cd) over Mo ratio (Cd/Mo) versus the product of cobalt (Co) and Mn (Co*Mn) can be used to determine paleohydrographic conditions during deposition of the Gordondale Member[61,64] (see discussion and Fig. S2 in the Supplementary Information). The Mo–U covariation trend for the Gordondale ORM suggests vigorous water-mass exchange with the open ocean[61]. The Cd/Mo–Co*Mn covariation corroborates this interpretation, suggesting an environment where upwelling of deep-water nutrients enhanced primary productivity[64]. Based on its local hydrographic regime and deposition from non-euxinic anoxic or euxinic bottom waters, the Gordondale Member in this core is ideal for estimating global ocean redox conditions before and during the T-OAE from redox-sensitive metal concentrations.

## Mass balance modeling of global seafloor total anoxia and euxinia

Modern seawater Re and Mo concentrations are regulated primarily by rivers, with minor seafloor hydrothermal inputs[33], and burial into sediments deposited from oxic, dysoxic/suboxidizing, and anoxic (Re) or euxinic (Mo) bottom waters[18,19]. A recent compilation of sedimentary Re data for modern marine environments showed that Re is removed most efficiently to the anoxic sediment sink as indicated by an average burial rate of 1.3 ng cm$^{-2}$ yr$^{-1}$, which is on average three factors greater than dysoxic/suboxidizing ($4.2 \times 10^{-1}$ ng cm$^{-2}$ yr$^{-1}$) and three orders of magnitude greater than oxic ($1.6 \times 10^{-3}$ ng cm$^{-2}$ yr$^{-1}$) settings[18]. The dysoxic and suboxidizing settings were combined to enable model estimates for the extent of seafloor anoxia. Burial rates of Re into sediments beneath non-euxinic anoxic and euxinic bottom waters are not significantly different because Re removal to organic-rich sediments depends on $O_2$ deficiency rather than $H_2S$ availability[18,36,37]. In contrast, Mo removal to these sediments hinges on the presence of free $H_2S$[35,56] such that there is a large offset between average Mo burial rates in oxic ($2.75 \times 10^{-3}$ µg cm$^{-2}$ yr$^{-1}$) and dysoxic/suboxidizing/non-euxinic anoxic (0.27 µg cm$^{-2}$ yr$^{-1}$) sediments versus euxinic (1.53 µg cm$^{-2}$ yr$^{-1}$) settings[19]. The high Re and Mo burial rates in anoxic and euxinic settings, respectively, means that a relatively minor expansion of anoxic or euxinic seafloor will result in substantial drawdown of these metals from seawater until a new steady state is reached. A smaller oceanic metal reservoir should lead to lower enrichments in marine ORM deposited from locally non-euxinic anoxic or euxinic waters. Hence, Re and Mo enrichments in ORM, coupled with a recently developed mass balance model[18] can be used to track changes in the extent of global ocean anoxia and euxinia associated with the T-OAE.

First-order approximations of global anoxic and euxinic seafloor area can be inferred from Re$_{auth}$ and Mo$_{auth}$ enrichments in ORM deposited in areas well-connected to global ocean circulation[18,19]. The dissolved Re and Mo seawater reservoirs are regulated by a mass balance accounting for metal input and output fluxes. Anoxic or euxinic sediments represent the output with the largest burial rate for Re and Mo, respectively. Thus, authigenic concentrations scale inversely with the areas of anoxic or euxinic sediment deposition in the whole ocean by Eq. 1 (after ref. [18]):

$$X_{\text{auth}} = \left(\frac{1}{\text{BMAR}}\right)\left(\frac{b_{\text{a,e}}^{\text{M}\,2}}{b_{\text{a,e}}^{\text{C}}}\right)\left(\frac{F_{\text{in}}}{\sum A_x b_x}\right) \quad (1)$$

where $X_{\text{auth}}$ is the authigenic concentration of the trace metal in locally anoxic (Re) or euxinic (Mo) sediments, BMAR is the local bulk sediment mass accumulation rate, $b_{\text{a,e}}$ is the metal burial rate in anoxic (a) or euxinic (e) environments (superscript M indicates the modern $b_{\text{a,e}}$ value, and superscript C indicates the $b_{\text{a,e}}$ derived after each model iteration and is dependent on assumed depth of anoxia and the

associated organic carbon burial; see Supplementary Information for full definition), $F_{\text{in}}$ is the metal's annual riverine input flux, $A_x$ is the seafloor area covered by sediment sink $x$ (where $x$ is the oxic, dysoxic/suboxidizing, or anoxic/euxinic areas) and $b_x$ is the characteristic metal burial rate in each sink environment. To estimate past areas of total seafloor anoxia or euxinia, the above equation is solved iteratively with $A_{\text{a,e}}$ from 0 to 100% of the total seafloor area. The full derivation (Equations S2–S8) and constants for the system of equations (Table S1) are provided in the Supplementary Information and applied in the Source Code file.

Application of the mass balance model to data from the Gordondale Member before and during the T-OAE requires samples selected from locally anoxic (Re) or euxinic (Mo) ORM[18,19]. Thus, the euxinic seafloor area from the Mo mass balance is not calculated for the locally suboxidizing/non-euxinic anoxic MI-2 and MI-3 or more oxygenated Post-N-CIE interval, but the anoxic seafloor area can be deduced by the Re model from all Pre-N-CIE and N-CIE intervals, allowing an assessment of global ocean redox changes within these intervals. To produce a representative model of the change in global anoxic and euxinic seafloor area before and during the T-OAE, local spatiotemporal parameters were applied to these stratigraphic intervals. We assess the impact on mass balance model solutions by the BMAR (Equation S9, Table S2), thermal maturity (Equation S10, Table S3, Fig. S3), sample dataset filtering protocols (Table S4) and Early Jurassic Re and Mo input fluxes from rivers and hydrothermal sources (Tables S5 and S6). We present an environmentally realistic scenario, with further sensitivity analysis described in the Supplementary Information.

The modern Re and Mo riverine fluxes to the oceans are $4.29 \times 10^5$ mol yr$^{-1}$ and $3.00 \times 10^8$ mol yr$^{-1}$, respectively[18,33], which is the assumed baseline flux prior to the N-CIE given broadly similar atmospheric $O_2$ levels at that time compared to modern[65]. Riverine fluxes scale with the rate of continental weathering, which is estimated to have increased by 215–530% over a 100–200 kyr period at the onset of the T-OAE based on Os and calcium isotope data[11,12]. The Os isotope data is from the Red Deer Member[11], which is closely related to the Gordondale Member in the c-B6-A/94-B-8 core, however, a sill between the units may have created minor basin restriction during Red Deer Member deposition[66]. The Gordondale Member in the c-B6-A/94-B-8 core contains consistently elevated authigenic Re ($208 \pm 91$ ng g$^{-1}$, 1 s.d.) and total organic carbon (TOC) content ($6.4 \pm 2.2\%$, 1 s.d.), while the East Tributary section hosting the Red Deer Member contains variably low Re concentrations ($52 \pm 49$ ng g$^{-1}$, 1 s.d.)[11] and moderate TOC content ($3.9 \pm 1.0\%$, 1 s.d.)[40]. The lower Re concentrations at East Tributary are not a result of more oxygenated conditions because sedimentary Fe speciation data indicates deposition from locally euxinic bottom waters[27]. Instead, minor hydrographic restriction at the Red Deer Member depositional locality could have caused lower Re concentrations at this locality. Consequently, local seawater $^{187}$Os/$^{188}$Os may be higher than global seawater such that the weathering rate evaluation (215–530% increase) from the East Tributary section[11] may be a mild overestimation. Hence, we apply a conservative threefold increase of Re and Mo riverine flux at the onset of the N-CIE and maintain this flux through both MI-3 and MI-4.

Hydrothermal Re and Mo fluxes to modern seawater are poorly constrained but are likely minor compared to riverine flux[18,33]. Sensitivity analyses of hydrothermal fluxes demonstrate that its subordinate contribution compared to river flux obviate its impact on the Re and Mo mass balance (see Supplementary Information), thus the hydrothermal flux was excluded from the mass balance model.

## Maximum extent of anoxia at the onset of the T-OAE

To evaluate the global seafloor extent of anoxia and euxinia during the T-OAE, the Re$_{auth}$ and Mo$_{auth}$ from the defined chemostratigraphic intervals of the study section are compared to modeled metal

enrichments for given seafloor sink areas (Fig. 3). The euxinic MI-1 features high $Re_{auth}$ and $Mo_{auth}$ of $215 \pm 88$ ng g$^{-1}$ and $168 \pm 91$ µg g$^{-1}$, respectively (1 s.d.). Such large enrichments qualitatively suggest a large dissolved inventory of these metals in the ocean, and thus small areas covered by anoxic or euxinic seafloor. This is confirmed quantitatively by the Re and Mo models which yield independently consistent total global seafloor areas covered by general anoxia (0.14–2.5%; minimum and maximum derived from ± 1 s.d. on the mean of the authigenic enrichments) and euxinia (0.18–1.9%). Euxinia is a component of the total anoxic seafloor area, so this overlap between the two global settings suggests that non-euxinic anoxia was a minor component of the anoxic seafloor, and euxinia dominated as in modern anoxic environments. The overlying MI-2 and MI-3 are locally anoxic but non-euxinic, so only the Re model is applied. These intervals have $Re_{auth}$ of $125 \pm 68$ ng g$^{-1}$ and $192 \pm 95$ ng g$^{-1}$ (1 s.d.), respectively, when dysoxic samples are removed based on the $Re/Mo_{auth}$ (Fig. 2a). The riverine flux parameter is altered from lower values in the Pre-N-CIE interval to higher values associated with increased continental weathering during the T-OAE in the N-CIE interval. The model results for these intervals yield total anoxic seafloor areas of 1.1–8.1% and 3.9–100%, respectively. We cannot place constraints on the proportion of non-euxinic versus euxinic seafloor within the total anoxic seafloor estimates we have provided here, as the Mo model is not reliable when local conditions are non-euxinic. The euxinic MI-4 has the highest $Re_{auth}$ of $277 \pm 74$ ng g$^{-1}$ and elevated $Mo_{auth}$ of $108 \pm 22$ µg g$^{-1}$, which correspond to a total anoxic seafloor area of 2.9–6.4% and euxinic seafloor area of 3.9–6.2%, respectively, when riverine flux is held at three times modern in the Re and Mo models. Like the MI-1, the MI-4 areas of total anoxic and euxinic seafloor are relatively consistent, which indicates that anoxic settings were likely dominated by euxinia. This independent consistency between the results from the Re and Mo models for those two intervals ($p > 0.16$; paired t-tests) also highlights the validity of the model calibration, both in the original studies (system of equations, sink burial rates, initial conditions)[18,19] and in this study (local BMAR, input flux variation).

The large uncertainty in the MI-3 interval is a function of the asymptotic nature of the model when anoxia is greater than 10% total seafloor area, i.e., small variation in $Re_{auth}$ corresponds to large variation in estimated anoxic seafloor area. Additionally, variation in total anoxic and euxinic seafloor area for all intervals is rooted in the variance in authigenic enrichments for each interval, which likely reflects, at least in part, changes in local depositional conditions such as fluctuating redox conditions (e.g., short-lived, transient bottom water re-oxygenation events that are not readily detected using our local redox proxies), host phase availability/composition or sedimentation rates[67], rather than global redox changes. It is important to note that lower $Re_{auth}$ and $Mo_{auth}$ caused by local depositional factors do not actually convey meaningful information about the global extent of ocean anoxia. Hence, mean $Re_{auth}$ and $Mo_{auth}$ are a more robust global redox indicator and were used to calculate mean global anoxic seafloor areas of 0.84% (MI-1), 2.6% (MI-2), 6.9% (MI-3) and 4.1% (MI-4), and global euxinic seafloor areas of 0.47% (MI-1) and 4.8% (MI-4) of the total seafloor (Fig. 3). These estimates may reflect maximum constraints for the extent of seafloor anoxia and euxinia because $Re_{auth}$ and $Mo_{auth}$ higher than the means suggest lower extents of seafloor anoxia and euxinia.

The Pre-N-CIE interval exhibits global ocean redox conditions similar to modern, however, a minor expansion of seafloor anoxia directly preceding the N-CIE suggests that the T-OAE began prior to the classically recognized δ13C expression of the event. A similar observation was noted in the Tl isotope record from the East Tributary (Canada) T-OAE section and was suggested as a response to an initial pulse of LIP activity[27]. This early expansion was not documented in global redox reconstructions from Europe because pre-N-CIE bottom water conditions recorded in those sections are thought to have been generally oxic[20,21]. Our Re model suggest a maximum extent of global anoxia at the onset of the N-CIE, which is mirrored in a pulse to less negative ε205Tl in the East Tributary and Dotternhausen (Germany) sections in tandem with a sharp transition to minimum δ13C$_{org}$ [27].

The Re model also suggests a contraction of global anoxia towards the end of the N-CIE in our studied section. The N-CIE in the Gordondale Member of our studied section is depicted as a sustained minimum in δ13C$_{org}$ around −30‰ and sharp shift back to more positive δ13C$_{org}$ into the overlying PCS. In contrast, the East Tributary and Dotternhausen sections record N-CIEs featuring a discrete shift to the most negative δ13C$_{org}$ followed by a protracted rising limb to less negative δ13C$_{org}$ [21,40]. It is likely that the difference between the morphologies of the δ13C$_{org}$ profile at each section represent local sedimentological process such as sedimentation rate, sea level changes, local redox conditions, and/or minor hiatuses[40]. We interpret that the N-CIE in the Gordondale Member reflects only the portion of minimum δ13C$_{org}$ captured by the other two sections, and that the rising limb occurs in the more oxygenated overlying PCS in this location. Thus, we can only compare the findings from our Re model with the ε205Tl proxy in the interval of minimum δ13C$_{org}$ recorded in the other sections because Re enrichments in dysoxic settings do not reflect global seawater changes. The ε205Tl in the East Tributary and Dotternhausen sections in the interval of minimum δ13C$_{org}$ decreases from the peak (ε205Tl ≈ −1) at the onset of the N-CIE to more negative values upsection (ε205Tl ≈ −4)[27] which, like our shift to greater Re enrichments through the N-CIE, suggests an increase in global oxic seafloor settings and a contraction of anoxia.

Molybdenum and S isotopes have indicated a global contraction of euxinic bottom waters only after the N-CIE[21,27,31]. However, the oceanic residence times of Re and Tl are shorter than Mo[32,33], equating to a faster response time for the global Re and Tl proxies. This can explain the apparently earlier contraction of general anoxia within the N-CIE, rather than following it. It is possible that euxinia responded in tandem with the contraction of general anoxia, however, the inability to estimate euxinic seafloor in MI-2 and MI-3 due to locally non-euxinic conditions, and the slower response time for changes to δ98Mo in studies of the European sections, does not allow for observation of the euxinic trend within the N-CIE.

The ε205Tl in the rising limb from the East Tributary and Dotternhausen sections show a second, protracted shift to less negative values suggesting an expansion of globally low-O$_2$ seafloor that extends beyond the end of the N-CIE[27]. The Post-N-CIE interval in the studied core is more oxygenated and thus not conducive for Re mass balance modeling, therefore, we cannot make a statement on the trajectory of global seafloor redox changes from the Re model beyond our MI-4.

The increase in seafloor area covered by anoxic conditions likely played a role in the extinctions that occurred during the T-OAE. The modern continental shelf and slope where biota is most heavily concentrated account for 3.6 and 5.6% of the global ocean area, respectively[68]. If the area of Toarcian continental shelf and slope were not substantially different from the present, then our model suggests that the maximum expansion of anoxia in the early portion of the T-OAE did not extend far into the deep ocean but remained largely within the continental margins (e.g., Panthalassa ocean margins)[40,43–46] or epicontinental seas (e.g., Tethys sea sub-basins)[20–25,41,42]. The Re mass-balance model results suggest that the open deep oceans away from continental margins likely remained predominantly oxygenated during the T-OAE, although potentially below modern levels[29,69].

Several studies have invoked anoxia, and specifically sulfidic anoxia, as a likely kill mechanism for marine fauna during this and other mass extinctions[70–72], or as a factor in ecological reorganization featuring a less diverse but more resilient set of marine fauna[73–75]. A global ammonite and foraminifera species analysis shows a steep decline in diversity at or near the onset of the T-OAE, followed by species recovery before the end of the event[2], mirroring the changes in

global ocean redox conditions observed here. Localized studies of various marine genera (e.g., corals in Moroccan and Italian sections[76]; benthic foraminifera from outcrop in Algeria and Portugal[77] and Tibet[78]) do not always exactly reflect our global redox trend likely due to species resolution in those studied sections, variable tolerance of low-$O_2$ conditions among species, and sensitivity to other perturbations. However, broad trends often show a significant decline in diversity through the T-OAE which may be reflective of the increase in global anoxic seafloor. Moreover, because our models encompass an averaged global redox landscape, local redox variations are not effectively captured. Thus, correlation between our global redox changes and local faunal patterns should not be relied on to infer anoxia as the sole cause of marine biodiversity decline and/or extinction, although on a global scale it was likely a contributing factor.

Our approach towards inferring global ocean redox conditions using Re and Mo enrichments deposited from locally anoxic and euxinic waters in open-ocean settings may be applicable to other intervals of the Phanerozoic characterized by severe mass extinctions, or in forecasting changes to the modern world. The modern Earth is experiencing surface changes such as an enhanced greenhouse atmosphere spurred by industrial, rather than LIP, activity. The resulting environmental feedbacks associated with the accumulation of anthropogenic greenhouse gases is leading to species loss akin to those of past mass extinction events[79]. Application of the coupled Re–Mo mass balance model may improve understanding of the magnitude of ocean redox changes during ancient expansions of oceanic anoxia and euxinia and their effects on biodiversity, and thus may enable prediction of the results of human impacts on the biosphere. The coupled Re–Mo paleoredox proxy may also help track longer-term changes in the distribution of euxinic versus non-euxinic seafloor throughout the Phanerozoic Eon.

## Methods

Permission for core sampling was granted by the British Columbia Oil and Gas Commission (BCOGC). Sampling reports can be accessed online using the BCOGC eLibrary and the Well Authorization # 31257. Remaining sample material is currently held at the University of Waterloo Metal Isotope Geochemistry Laboratory (MIGL).

### Trace metals

Sample preparation and analysis for trace metal content was performed at the MIGL, University of Waterloo. Samples without secondary features (e.g., fossils, veins, macroscopic pyrite nodules) were chosen for analysis. Samples were crushed without metal contact and powdered in a Retsch ball mill with agate grinding jars. Approximately 100 mg of powder was ashed overnight at 550 °C to oxidize organic matter. Ashed powders were digested at 110 °C in 2.5 ml concentrated $HNO_3$ and 0.5 ml concentrated HF for 48 h, followed by digestion in 4 ml *aqua regia* for 48 h and finally in 2 ml concentrated HCl for 24 h. Digested solutions were diluted 400-fold for Re and Cd, and 6000-fold for other elements by 2% $HNO_3$ and trace HF was added to ensure sample stability. Elemental concentrations were measured on an Agilent 8800 triple-quadrupole inductively coupled plasma mass spectrometer calibrated against standard solutions containing metal concentrations designed to be similar to the matrix of ORM. Internal standards Sc, Ge, In, and Bi were used to correct for instrument drift. Instrument accuracy was verified by analyzing United States Geological Survey ORM standards SBC-1[80] and SGR-1b[81] multiple times during each analytical session. Concentration reproducibility was typically better than 5%. All elemental data presented in this article are available in the Supplementary Dataset 1.

### Organic carbon isotopes

Organic C isotope analysis was performed at the Environmental Isotope Laboratory, University of Waterloo. Sample powders were subjected to two rounds of 10% HCl acidification at 50 °C to leach carbonate, followed by rinsing with NanoPure water to remove excess acid. Approximately 1–10 mg of leached powder was placed in a foil cup for analysis on a Costech Instruments 4010 Elemental Analyzer coupled to a Thermo-Finnigan Delta Plus XL continuous-flow isotope ratio mass spectrometer. Carbon isotope ratios are reported against international and in-house standard reference materials calibrated to Vienna PeeDee Belemnite. Analytical precision was ± 0.2‰. All C isotope data is presented in the Supplementary Dataset 1.

## Data availability

The elemental and carbon isotope data in this study are provided in the Supplementary Dataset 1 (spreadsheet).

## Code availability

Mathematical system of equations and models were modified from Sheen et al. (2018) and Reinhard et al. (2013). We have provided a summary of the model in our Supplementary Information, and a user-modifiable spreadsheet as a Source Code file.

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

## Acknowledgements

Sample collection and analyses were funded by a Mitacs Accelerate grant (IT10033, A.K. and B.K), NSERC Discovery Grants (RGPIN-2013-435930/RGPIN-2019-04090, B.K.), and the Canada Research Chairs program (B.K.). The Metal Isotope Geochemistry Laboratory was funded by the Canada Foundation for Innovation, Ontario Research Fund, and University of Waterloo.

## Author contributions

The study was developed by A.K. and B.K. Analysis and data processing were carried out by A.K. Preparation of the manuscript was undertaken by A.K., with input from B.K.

## Competing interests

The authors declare no competing interests.
