## [Peer Review File · Nature Communications]

Global ocean redox changes before and during the Toarcian Oceanic Anoxic EventReviewer #1 (Remarks to the Author):

Review of Kunert and Kendall: Euxinic versus ferruginous anoxia before and during the Toarcian Oceanic Anoxic Event.

The authors present new trace metal data from shales spanning the T-OAE in Canada. The data are modelled to calculate that <10% of the global seafloor became anoxic during the event, with a slightly lesser extent of euxinic seafloor.

This was an interesting read, but a difficult paper to judge for Nat. Comms. On the one hand, the dataset is detailed, and the importance of the subject is quite clear. On the other hand, I am not at all convinced by the interpretations of ferruginous conditions based on Re concentrations, and the conclusions don't significantly advance what we already know from Mo and Tl isotope data. That said, I do believe that scientific replication using different methods is very important, and in that regard the similarity between the conclusions reached here and those of other authors using different methods is important. In the event of a hung jury I would tend towards being more positive than negative.

My main comments are below, after which I have some line by line comments:

1. Ferruginous conditions. The authors make the argument that Re concentrations are tracing ferruginous conditions. As you will see from the line by line comments below, I don't think this is a robust interpretation. Re thiolates, just like Mo, and has been argued to co-precipitate with Mo in the presence of an Fe-S-Mo particle (Vorlicek et al., 2018; Helz, 2021). The interpretation that high [Re] and low [Mo] = ferruginous conditions is probably just as likely to represent sub-oxic conditions. Part of the problem here probably lies with the fact that the literature keeps trying to force geochemical behaviour into little redox silos (anoxic, sub-oxic etc) when the reality is far more complex and phase-specific.

2. Modelling: I have no problem with the maths behind the model, which appears robust. However, from a readability perspective, there is basic information in the SI that probably needs to be in the main text to explain how the model works. At present, the main text reads like a list of model boundary constraints, without a clear, qualitative description of the approach itself. This is important for the general readership of Nat. Comms.

Minor comments:

Line 28: Note variable estimates up to >1Myr by Kemp et al. (2011).

Line 31: When thinking about weathering during the T-OAE it would seem odd not to cite Cohen et al. (2004) who first proposed intense continental weathering using Os isotopes.

Line 87: Few, if any, of the studies cited here defined the C-isotope signature of the T-OAE. Look at early papers by Jenkyns (e.g. 1988, AJS, 1997 etc) and a recent paper in NoS by Erba et al. (2022/in press).

Line 99 is a bit circular/awkward wording. The age of the T-OAE should be the same everywhere – it isn't distinctly 182–185 Ma just in this core. Better to write that stratigraphic and age constraints from correlative sections indicate that the studied succession contains deposits of T-OAE age.

Line 106–112: I am not sure I agree with the reasoning here for identifying ferruginous conditions. Re and U are not uniquely buried under anoxic and S-limited conditions. U reduction occurs at relatively high redox couple conditions (near the Fe^{III}/II reduction boundary) and does not require complete anoxia to occur (see vast amounts of

literature on the subject). Re burial mechanisms aren't well understood but probably involve a degree of S speciation (see Helz and Dolor, 2013).

Line 118: Again, I disagree. If U and Re burial is more or less constant, with variable burial of Mo/V, isn't that most conservatively interpreted as constant sub-oxia (in sediments) with pulses of greater H₂S availability/stagnation driving Mo/V burial? I don't see how you rule out suboxic conditions with these data?

Line 132: Or a reduction in [H₂S] leading to a decrease in Mo burial, with suboxia remaining?

Line 137: The Mo/U co-variation is convincing, but the Cd/Mo isn't. This is because the entire Cd/Mo dataset cluster around the cutoff for productivity versus preservation-driven settings, and doesn't clearly indicate upwelling as argued. Do the author have Mn and Co data to produce a full Cd/Mo/Co/Mn cross-plot, which is more informative?

Section up to line 193: I think this section needs some additional explanation of how the model works. This is described quite loosely, with most details in the SI, but the casual reader will wonder how you have gone from local sedimentary concentrations to global seafloor. Are you assuming that burial fluxes in the new studied core are broadly indicative of burial fluxes in similar redox environments globally? If so, this is a useful bit of qualitative information to state.

Line 244–245: I get a bit frustrated reading statements like this. The T-OAE community has generally not considered 'global' or even 'widespread' ocean anoxia to mean 'the whole ocean' (or even the deep ocean) for decades. However, it is repeated rolled out as a counter-point to studies showing basinal deoxygenation. 99.9% of folks who work on this event would agree to the statement that the T-OAE was a general expansion of anoxia, best expressed in marginal basins but also seen in coastal locations (Q. Elizabeth Island, high Arctic, even Chinese Lakes), driven by some sort of common global-scale environmental driver. It reminds me of when radio or TV shows roll out the wizened climate sceptic to provide a counter point to a global warming discussion.

Line 256: Be careful using 'the Re-Mo paleoredox proxy' because this term has been used previously in reference to Re/Mo ratios (e.g. Crusius et al., 1996; Helz, 2021) whereas here I think the authors use it in relation to quantifying euxinic versus ferruginous redox conditions.

Reviewer #2 (Remarks to the Author):

This is a well written manuscript containing well-presented data that is important for the fields of paleoceanography, paleoredox proxy studies, and future predictions of climate change-caused oceanic deoxygenation. I suggest minor revisions listed below and recommend that the revised manuscript be published in Nature Communications.

Abstract: I am not sure if I would start with writing H₂S, Fe²⁺, and O₂ for sulfide, ferrous iron, and oxygen given that they weren't previously defined as such. However, I realize that the targeted reader most likely knows what is meant by this, so this is a slight suggestion to define those compounds before using that format but use your best judgement.

L22: "...7% of anoxic seafloor dominated by euxinia" could be misunderstood as the euxinic portion was only 7% of the anoxic seafloor area. Maybe reword this to be more clear by added "...7% of the anoxic seafloor, which was dominated..."

Main:

L53-54: "limited areas of oxic seafloor" makes this statement seem redundant, as if you are saying that oxic Mn oxide burial shifts the seawater Tl isotope composition lighter

and heavier. Consider changing to "whereas when oxic seafloor area is limited, seawater Tl isotope compositions become heavier" or "whereas increased anoxic seafloor area shifts seawater to heavier values".

L60: I found the term deconvolve to be a bit confusing, consider changing it to "distinguish" or a more common term.

L63 and 66: you say "rhenium" on line 63 and "Re" on line 66, I would change L63 to say "rhenium (Re)" so that by L66, Re has been defined.

L72-74: add reference(s) at the end of this statement.

L90 and 93: on L90 you write 3 permil to 4 permil, but on L93, you omit the permil on the first number, so I would just change one of these to make it consistent.

L102 and 104-105: Same thing as my suggestion for L63&66, define uranium as U and vanadium as V before using them as such.

L109: you use a hyphen for "bottom-waters", but in other places you don't, I suggest to pick one format and be consistent.

L110-111: you mention Re/Mo and U/Mo ratios for euxinic bottom waters but only Re/Mo ratios for ferruginous or suboxic waters. Is there a reason for excluding ferruginous and suboxic U/Mo ratios?

L133: add (Cd) after "cadmium"

L138-140: this sentence could be misunderstood, I would change it to "Based on its local" and take out the "its" before "deposition", just to make it clear that you are talking about the Gordondale member's regime & deposition because when I first read it I thought you were trying to say the regime's deposition which didn't make sense.

L229-230: I would change "Lower Pre-N-CIE and Upper.." to "Lower Pre-N-CIE and in the Upper.." because I can see this being misunderstood as the offset between these intervals rather than the offset between the anoxic and euxinic areal extents. I might also add the values of Anoxic and Euxinic in parentheses after "Lower Pre-N-CIE" and "Upper N-CIE".

L284: I would add "Vienna" in front of "Pee Dee Belemnite".

Supplementary Information:

Overall, I think the SI is very informative and explains any discrepancies of the main text very well. I just have a few suggestions:

L75: Is there a reason you use Mo/U ratio in the SI but the U/Mo ratio in the main paper?

When you say "seawater ratio" on Lines 75, 76, 79, and 83, do you mean the modern seawater Mo/U ratio, and if so, do you define this anywhere (not the value, just the meaning) because I found it a bit confusing.

L185: add "in"-> "...increase in continental weathering rate..."

L372: take out "to cause" -> "...may have been a higher..."

Reviewer #3 (Remarks to the Author):

General Comments:

Kunert and Kendall have presented a detailed trace metal (TM) and carbon isotope analysis of a single drill core from the Western Canada Sed Basin spanning the Early Jurassic T-OAE. The authors present a new mass balance model constraints of the extent of reducing seafloor extents based upon Re and Mo concentrations and ratios. These new constraints are consistent with previous estimates based upon Mo isotopes that suggested <10% of seafloor euxinia. These new data are of good quality and the mass balance approach I find novel. The authors in the title and bold paragraph invoke ferruginous conditions playing an important part of this manuscript however in the main discussion of data and modeling section I don't see the tie into anoxic conditions where free-Fe²⁺ is within the water column and how exactly their proxies point to this particular portion of the marine redox ladder (i.e., Fe reduction). Seems as though their TM's also are responsive under the Mn-reduction and Nitrate-reduction windows of the marine redox ladder (Froelich et al 1979; Algeo & Maynard, 2004; Owens, 2019 and refs therein). Overall I feel like there could be better integration with previous global redox proxy datasets, the authors do have a nice preliminary discussion on the Them et

al TI isotope datasets in their introduction, however there is little to no integration of that global redox proxy data with their own. How do these proxy results compare over the same time interval? How do they potentially inform about expected changes in the marine redox ladder as the Early Jurassic oceans were purportedly getting more reducing? Do we see the leads and lags in proxies in the expected ways? There is no mention of CAS-S34 work done by Gill et al 2011, this is another global redox proxy for euxinic conditions, seems like this would be relevant to this paper's discussion. Overall I don't know if I understand the bigger picture results presented herein in the context of other global redox datasets that have been published for the T-OAE interval. I see the model constraints are novel but would like to know why these new constraints are critical to our understanding of this interval in Earth history.

Specific Comments:

Lines 13-16: Okay so no modeled estimates of anoxia or euxinia exist but reducing conditions have been invoked to be widespread for some time now (Gill et al. 2011; Bowden et al., 2006; Farrimond et al., 1989, 1994; Pancost et al., 2004; Raiswell and Berner, 1985; Raiswell et al., 1993; Schouten et al., 2000; van Breugel et al., 2006; Them et al 2018). What is the big take away with these new modeled estimates (Re & Mo concentrations/ratios) which overlap with Mo-isotope estimates???

Line 31: marine or oceanic productivity

Lines 95-100 & SI information on correlation and age constraints:

The correlation of this drill core with radiometric ages or biostratigraphy seems largely based upon GR-data and logs that don't look that conclusive in the Figure S1. This drill core is > 500km away from a section with good ammonite biostrat and near 1000km away from a section with U-Pb ages.

Seems like the age assignments should be dashed in at the very least given the distance to sections with age constraints. Additionally, the magnitude of C13 change is notably smaller than the other sections with good radiometric or biostrat control and thus it's hard to determine for certainty that this is the N-CIE associated with the T-OAE considering all of the other sections from this region and basin have larger magnitudes and there are other smaller CIE's known in the Early Jurassic (e.g., Cramer and Jarvis, 2020; GTS). The authors are most likely right in their age assignments but seems like they need to call out the caveats a bit more clearly in the main text.

Lines 121-128:

These named intervals are very confusing and hard to keep track of in a paper that already has many acronym's to keep track of. I am unsure of the purpose of even naming these intervals within the Gordondale member. I think the authors should reconsider the purpose of these named intervals as these seem to only be applicable to this drill core with these specific set of TM datasets.

Instead the authors could just refer their TM trends relative to strata or just have a simplified approach with two labels of pre-NCIE and NCIE.

Lines 205-207:

which stratigraphic intervals are you referring to here? Hard to follow this.

Figure 2:

Why are these lower and upper boundaries of preNCIE and NCIE chosen? They seem to be right at the geochemical breaks in the TM dataset the authors have presented. Are these official stratigraphic designations that others use? If not then need a sentence or two defining these L & U intervals.

Not sure I would call the end of the NCIE at 1575m as this is only based up on one data

point where C13 varies by less than 1‰ between the underlying data point. This could be the end of the NCIE but could also be an outlier of these bulk C13org data.

how are suboxic and anoxic defined and differentiated?

Figure3:

Would recommend rearranging these plots so they are in stratigraphic (i.e., time order) with "pre-NCIE" model results first then "NCIE" results second. This will make things flow more logically with text discussion.

Would recommend adding a note here in the caption as to why no "Upper pre-NCIE" and "Lower NCIE" model runs. As you have to refer back to figure 2 to figure this out...i.e., No enrichments in [Mo] through these intervals.

Dear Reviewers,

Thank you for your constructive feedback. We have taken these into account during the revision. Revisions are highlighted using the “Track Changes” function in the submitted manuscript document. A PDF without mark up is also provided to facilitate reading if preferred. Only a document with mark up is provided for the supplementary file.

Below, find our responses to the major concerns summarized by the editor, and point-by-point responses to all reviewer comments. Line numbers are indicated for minor changes (based on line number in the no-markup PDF).

MAJOR COMMENTS

Fully discuss where your work fits in with previous studies and proxy datasets.

[Reviewer #3] Overall, I feel like there could be better integration with previous global redox proxy datasets, the authors do have a nice preliminary discussion on the Them et al TI isotope datasets in their introduction, however there is little to no integration of that global redox proxy data with their own. How do these proxy results compare over the same time interval? How do they potentially inform about expected changes in the marine redox ladder as the Early Jurassic oceans were purportedly getting more reducing? Do we see the leads and lags in proxies in the expected ways? There is no mention of CAS-S34 work done by Gill et al 2011, this is another global redox proxy for euxinic conditions, seems like this would be relevant to this paper's discussion. Overall, I don't know if I understand the bigger picture results presented herein in the context of other global redox datasets that have been published for the T-OAE interval. I see the model constraints are novel but would like to know why these new constraints are critical to our understanding of this interval in Earth history.

[Reviewer #3] Lines 13-16: Okay so no modeled estimates of anoxia or euxinia exist but reducing conditions have been invoked to be widespread for some time now (Gill et al. 2011, Bowden et al., 2006; Farrimond et al., 1989, 1994; Pancost et al., 2004; Raiswell and Berner, 1985; Raiswell et al., 1993; Schouten et al., 2000; van Breugel et al., 2006; Them et al 2018). What is the big take away with these new modeled estimates (Re & Mo concentrations/ratios) which overlap with Mo-isotope estimates???

Mentioned in Abstract and discussed in section “Maximum Extent of Anoxia at the Onset of the T-OAE”.

We have made extensive additions to the interpretations of our model results based on the reviewer comments. These are summarized below:

1. Ocean residence times: Previous studies have used Mo, S, and TI isotopes to infer global redox conditions of the Early Jurassic. Mo and S have longer oceanic residence times than Re, and TI has a much shorter residence time. We argue that Re (which we use to model our entire Gerdondale interval) has a “goldilocks” residence time with respect to these other elements. That is, Mo and S have longer residence times, thus cannot respond as rapidly to ocean redox changes, and TI is much shorter, thus may fall victim to local basinal conditions especially in restricted environments.
2. Differences between Mo & S isotope and Re models: We see a later contraction (after the N-CIE) of euxinia based on Mo and S isotope models compared to our Re model (before the end of the N-CIE). This is likely due to the shorter residence time of Re, hence why we observe an earlier contraction of anoxia in the Re model.
3. Similarities and differences between TI isotope and Re models: Both should respond fairly quickly to changes in redox conditions due to shorter ocean residence times. We observe an expansion of anoxia before the N-CIE in both TI isotope (Them et al. 2018) and Re models and a maximum extent of anoxic conditions at the onset of the N-CIE. Them et al. (2018). The TI isotopes in Them et al (2018) are discussed in a global

context well beyond the termination of the N-CIE as the local redox environment of those sections likely allowed for TI isotopes to record seawater compositions. However, in our studied section, this post N-CIE interval cannot be discussed due to locally dysoxic/oxic conditions (thus the Re model is not applicable). We compare the morphologies of the N-CIEs at all locations to attempt to compare the TI isotope and Re model trends through the majority of the N-CIE.

4. Comparison to global biodiversity trends: Because the Re model can provide a more detailed temporal trend than the Mo or S isotope models, it is possible to resolve global redox changes within the T-OAE interval. We compare the ocean redox trends we quantified to global biodiversity trends modelled by Caruthers et al. (2013). The redox and biodiversity models mirror each other in that the point of lowest species diversity occurs at or near the time of greatest seafloor anoxic area, followed by species recovery towards the end of the T-OAE in tandem with the contraction of anoxic seafloor. We discuss the use of our global ocean redox model to implicate anoxia as a factor in global biodiversity loss/extinction in the Toarcian.

Insert necessary caveats on your age constraints.

[Reviewer #1] Line 99 is a bit circular/awkward wording. The age of the T-OAE should be the same everywhere – it isn't distinctly 182–185 Ma just in this core. Better to write that stratigraphic and age constraints from correlative sections indicate that the studied succession contains deposits of T-OAE age.

[Reviewer #3] Lines 95-100 & SI information on correlation and age constraints: The correlation of this drill core with radiometric ages or biostratigraphy seems largely based upon GR-data and logs that don't look that conclusive in the Figure S1. This drill core is > 500km away from a section with good ammonite biostrat and near 1000km away from a section with U-Pb ages. Seems like the age assignments should be dashed in at the very least given the distance to sections with age constraints. Additionally, the magnitude of ^{13}C change is notably smaller than the other sections with good radiometric or biostrat control and thus its hard to determine for certainty that this in the N-CIE associated with the T-OAE considering all the other sections from this region and basin have larger magnitudes and there are other smaller CIE's known in the Early Jurassic (e.g., Cramer and Jarvis, 2020; GTS). The authors are most likely right in their age assignments but seems like they need to call out the caveats a bit more clearly in the main text.

Main text section "Local paleoenvironmental setting of the T-OAE in British Columbia" and in Supplementary Information.

We believe that the detailed explanations of age caveats are best left in the SI, however, as the reviewer suggests, we mention the spatial limitations on our correlations (sections with ammonite and geochronology ages located ~200 to 650 km from our studied section) in the main text. We have also revised the wording in the main text to use less definitive language, e.g., "...suggests the section contains Toarcian deposits" rather than "constrains the age to between ~185 and 182 Ma". We also state explicitly that the correlations are made using gamma ray logs in addition to the $\delta^{13}\text{C}_{\text{org}}$ data from those cores/outcrop sections.

In addition, the original Re-Os ages cited were from an Alberta Geological Survey Report (Pana et al. 2018), but updated ages were provided in Toma et al. (2020, *Palaeo3*) for the original Re-Os core, as well as two additional cores (one of which features the $\delta^{13}\text{C}_{\text{org}}$ profile from Them et al. 2019). These new data have been appended to Figure S1 and discussed in the revised version of the SI.

Finally, Reviewer 3 mentioned that one cited core location was almost 1000 km from the studied core. This is not true, but we see how the original Figure S1 may have been easily misread, as the studied core was placed second-from-left in the cross section. This has

been amended so that the studied core is now the first stratigraphic profile. A small study area map was also added to the figure to provide a general spatial scale and rough locations of all cores in the cross section.

Very thoroughly address Reviewer 1's comments on your interpretation of ferruginous conditions from Re concentrations.

[Reviewer #1] Ferruginous conditions. The authors make the argument that Re concentrations are tracing ferruginous conditions. As you will see from the line-by-line comments below, I don't think this is a robust interpretation. Re thiolates, just like Mo, and has been argued to co-precipitate with Mo in the presence of an Fe-S-Mo particle (Vorlicek et al., 2018; Helz, 2021). The interpretation that high [Re] and low [Mo] = ferruginous conditions is probably just as likely to represent suboxic conditions. Part of the problem here probably lies with the fact that the literature keeps trying to force geochemical behaviour into little redox silos (anoxic, suboxic etc.) when the reality is far more complex and phase specific.

[Reviewer #1] Line 106–112: I am not sure I agree with the reasoning here for identifying ferruginous conditions. Re and U are not uniquely buried under anoxic and S-limited conditions. U reduction occurs at relatively high redox couple conditions (near the Fe(III)/Fe(II) reduction boundary) and does not require complete anoxia to occur (see vast amounts of literature on the subject). Re burial mechanisms aren't well understood but probably involve a degree of S speciation (see Helz and Dolor, 2013).

[Reviewer #3] The authors in the title and bold paragraph invoke ferruginous conditions playing an important part of this manuscript however in the main discussion of data and modeling section I don't see the tie into anoxic conditions where free-Fe²⁺ is within the water column and how exactly their proxies point to this particular portion of the marine redox ladder (i.e., Fe reduction). Seems as though their TMs also are responsive under the Mn-reduction and Nitrate-reduction windows of the marine redox ladder (Froelich et al 1979; Algeo & Maynard, 2004; Owens, 2019 and refs therein).

Local redox discussed in "Local paleoenvironmental setting of the T-OAE in British Columbia". Global use of Re and Mo models in section "Mass balance Modeling of Global Seafloor Total Anoxia and Euxinia".

We originally used the term "ferruginous" to describe the environment where bottom waters are anoxic, but do not contain H₂S. This definition is common in many paleoredox studies which split anoxia into ferruginous (i.e., non-euxinic) and euxinic (containing H₂S). However, we agree with the reviewers that the term can be easily misinterpreted, especially by a general readership, as it implies the accumulation of reduced Fe. We have therefore altered all instances of the use of "ferruginous anoxia" in our manuscript to "non-euxinic anoxia" which is a more general term and does not require the implied presence of reduced Fe. Our overall redox classification follows that of Algeo and Li (2020)/Algeo and Liu (2020) and conditions are explicitly defined within the main text.

The reviewers also question our use of Re as a tracer of ferruginous (non-euxinic) conditions, although it is unclear whether they are referring to its use as a local or global redox proxy. To alleviate confusion between the use of Re (and Mo) as local vs global redox proxies, we have completed a major revision of the section "Local paleoenvironmental setting of the T-OAE in British Columbia", shifting away from the use of absolute trace metal concentrations to infer local paleoredox conditions and only use the authigenic Re and Mo concentrations in the global mass balance models. For local interpretations, we now favour the use of trace metal ratios (Mo/U, Re/Mo) and comparison of Re, Mo, U, and V enrichments normalized to Al to determine whether the local depositional environment was oxic, dysoxic, suboxidizing, or anoxic (non-euxinic or euxinic) (after Algeo and Li, 2020/Algeo and Liu, 2020, and Bennett and Canfield, 2020).

Identifying local depositional redox conditions is necessary to identify stratigraphic intervals from which our global mass balance models can be applied. To clarify the goal of the local redox analysis, we have added a line of reasoning for evaluating local conditions for global reconstructions to the second paragraph of the section “*Local paleoenvironmental setting of the T-OAE in British Columbia*”.

For the global Re mass balance, sediments buried under locally anoxic conditions where enrichment “does not depend on H₂S availability in the water column [...] and allows us to merge [non-euxinic and euxinic] anoxia into one sink” (Sheen et al. 2018) are identified using Re/Mo_{auth}, Mo/U_{auth}, and V/Al vs. Mo/Al and V/Al vs. U/Al covariations (new addition to article with trends from Bennett and Canfield, 2020). Review of the compiled data in Bennett and Canfield (2020) also shows that suboxidizing (< 5 μM O₂) and fully anoxic settings support the highest Re enrichments, so suboxidizing settings identified in our local reconstruction are also selected for use in the Re model. Thus, we believe local redox interpretations for the Gordondale Member are robust and likely reflect at least generally suboxidizing or anoxic conditions where the Re model can be applied. Only two intervals (*Lower Pre N-CIE* and *Upper N-CIE*) are most likely euxinic where the Mo model is applied.

Move necessary modelling information from the SI into the methods/main text (where applicable).

[Reviewer #1] Modelling: I have no problem with the maths behind the model, which appears robust. However, from a readability perspective, there is basic information in the SI that probably needs to be in the main text to explain how the model works. At present, the main text reads like a list of model boundary constraints, without a clear, qualitative description of the approach itself. This is important for the general readership of Nat. Comms.

[Reviewer #1] Section up to line 193: I think this section needs some additional explanation of how the model works. This is described quite loosely, with most details in the SI, but the casual reader will wonder how you have gone from local sedimentary concentrations to global seafloor. Are you assuming that burial fluxes in the new studied core are broadly indicative of burial fluxes in similar redox environments globally? If so, this is a useful bit of qualitative information to state.

Section “Mass Balance Modeling of Global Seafloor Total Anoxia and Euxinia”.

We agree that we overlooked stating a qualitative outline of the model logic in the main text. We have added a brief description of the model logic in the section “*Mass Balance Modeling of Global Seafloor Total Anoxia and Euxinia*” in the main text. This outlines the trajectory from local redox conditions of organic-rich mudstone deposition to global areas of anoxia/euxinia as follows (as sentences in the text, rather than enumerated):

1. When ORM are deposited under locally anoxic (Re; both non-euxinic or euxinic anoxic) or euxinic (Mo) conditions, metal enrichments in sediments reflect the magnitude of the global seawater Re or Mo reservoir.
2. The dissolved Re and Mo seawater reservoirs are regulated by a mass balance accounting for metal input and output fluxes, and the major outputs of these metals are burial in anoxic or euxinic sediments.
3. Thus, enrichments (as authigenic concentrations) scale inversely with the areas of anoxic or euxinic sediment deposition in the whole ocean.

We have also added a summarized equation of the mass balance model, which illustrates the model parameters and the nature of the inverse relationship between authigenic enrichments and anoxic/euxinic seafloor areas. The reader is still directed to the SI for full modelling description/definitions.

MINOR REVISIONS (MAIN TEXT)

Reviewer 1

[Reviewer #1] Line 28: Note variable estimates up to >1 Myr by Kemp et al. (2011).

[L31] “The 300–500 kyr event^{3–5} coincides with...” changed to “The event, lasting an estimated 300 kyr to >1 Myr^{3–6}, coincides with...” (cited Kemp et al. 2011).

[Reviewer #1] Line 31: When thinking about weathering during the T-OAE it would seem odd not to cite Cohen et al. (2004) who first proposed intense continental weathering using Os isotopes.

[L34] Added the Cohen et al. (2004) citation + reference.

[Reviewer #1] Line 87: Few, if any, of the studies cited here defined the C-isotope signature of the T-OAE. Look at early papers by Jenkyns (e.g., 1988, AJS, 1997 etc.) and a recent paper in NoS by Erba et al. (2022/in press).

[L112]

We cited these seven articles for two reasons:

- 1) All state, in one way or another, that the T-OAE is defined by a broad positive excursion interrupted by a negative shift in the centre.
- 2) These represent the global reproducibility of the above trend during the T-OAE.

However, we do agree that Hugh Jenkyns’ work is foundational to the study of the T-OAE, therefore we have added the 1988 AJS paper, as well as the 2010 G3 Mesozoic OAE update.

[Reviewer #1] Line 118: Again, I disagree. If U and Re burial is more or less constant, with variable burial of Mo/V, isn't that most conservatively interpreted as constant suboxia (in sediments) with pulses of greater H₂S availability/stagnation driving Mo/V burial? I don't see how you rule out suboxic conditions with these data? AND Line 132: Or a reduction in [H₂S] leading to a decrease in Mo burial, with suboxia remaining?

[L204-205] Original statement was “...indicators reveal that deposition occurred under both ferruginous and euxinic bottom water conditions, and rarely under suboxic conditions.” As with other statements using the term “ferruginous” this has been revised to “non-euxinic anoxia”, and with the major revision to the local redox section, we have now updated the interpretations to show that deposition likely occurred under predominantly suboxidizing, non-euxinic anoxic, or euxinic conditions. We discuss how Re enrichments in modern settings (after Bennett and Canfield, 2020) occur equally in all three of these settings. See summary of local interpretations for the study section from L197-L216.

[Reviewer #1] Line 244–245: I get a bit frustrated reading statements like this. The T-OAE community has generally not considered ‘global’ or even ‘widespread’ ocean anoxia to mean ‘the whole ocean’ (or even the deep ocean) for decades. However, it is repeated rolled out as a counterpoint to studies showing basinal deoxygenation. 99.9% of folks who work on this event would agree to the statement that the T-OAE was a general expansion of anoxia, best expressed in marginal basins but also seen in coastal locations (Q. Elizabeth Island, high Arctic, even Chinese Lakes), driven by some sort of common global-scale environmental driver. It reminds me of when radio or TV shows roll out the wizened climate sceptic to provide a counter point to a global warming discussion.

This statement was removed as it did not add to the flow or conclusions of the manuscript.

[Reviewer #1] Line 256: Be careful using 'the Re-Mo paleoredox proxy' because this term has been used previously in reference to Re/Mo ratios (e.g., Crusius et al., 1996; Helz, 2021) whereas here I think the authors use it in relation to quantifying euxinic versus ferruginous redox conditions.

[L438] Changed to "the coupled Re-Mo mass balance model".

Reviewer 2

[Reviewer #2] Abstract: I am not sure if I would start with writing H₂S, Fe²⁺, and O₂ for sulfide, ferrous iron, and oxygen given that they weren't previously defined as such. However, I realize that the targeted reader most likely knows what is meant by this, so this is a slight suggestion to define those compounds before using that format but use your best judgement.

[Abstract] Removed Fe²⁺ due to other revisions (i.e., removal of the term "ferruginous"), changed "H₂S-rich" to "sulfide-rich". Left "O₂", as it should be understood by readers and changing to "oxygen" could imply other forms of oxygen-bearing molecules.

[Reviewer #2] L22: "...7% of anoxic seafloor dominated by euxinia" could be misunderstood as the euxinic portion was only 7% of the anoxic seafloor area. Maybe reword this to be more clear by added "...7% of the anoxic seafloor, which was dominated..."

[L21-22] Corrected to "...up to ~7% total seafloor anoxia, which was dominated by euxinia."

[Reviewer #2] L53-54: "limited areas of oxic seafloor" makes this statement seem redundant, as if you are saying that oxic Mn oxide burial shifts the seawater Tl isotope composition lighter and heavier. Consider changing to "whereas when oxic seafloor area is limited, seawater Tl isotope compositions become heavier" or "whereas increased anoxic seafloor area shifts seawater to heavier values".

[L59] Went with the latter statement, "whereas increased anoxic seafloor area will shift seawater $\epsilon^{205}\text{Tl}$ to heavier values."

[Reviewer #2] L60: I found the term deconvolve to be a bit confusing, consider changing it to "distinguish" or a more common term.

[L66] Changed to "distinguish" as suggested.

[Reviewer #2] L63 and 66: you say "rhenium" on line 63 and "Re" on line 66, I would change L63 to say "rhenium (Re)" so that by L66, Re has been define.

[L71] Agreed. This and all further instances have been modified in the text to define the name/abbreviation of an element at its first mention. For further comments, we have noted "Corrected".

[Reviewer #2] L72-74: add reference(s) at the end of this statement.

[L92-94] Cited Erickson and Helz (2000) for "Mo preferentially forms thiomolybdate complexes in the presence of dissolved H₂S", and Reinhard et al. (2013) for "significantly elevated [Mo burial efficiency in euxinic settings] compared to ... non-euxinic settings."

[Reviewer #2] L90 and 93: on L90 you write 3 permil to 4 permil, but on L93, you omit the permil on the first number, so I would just change one of these to make it consistent.

[L115 & L118] Added a permil sign to the second instance, so both listed numbers are followed by the permil sign.

[Reviewer #2] L102 and 104-105: Same thing as my suggestion for L63&66, define uranium as U and vanadium as V before using them as such.

Corrected.

[Reviewer #2] L109: you use a hyphen for "bottom-waters", but in other places you don't, I suggest to pick one format and be consistent.

Removed hyphen from all instances. Now reads "bottom waters".

[Reviewer #2] L110-111: you mention Re/Mo and U/Mo ratios for euxinic bottom waters but only Re/Mo ratios for ferruginous or suboxic waters. Is there a reason for excluding ferruginous and suboxic U/Mo ratios?

The Mo/U ratios (no longer U/Mo, for consistency) for anoxic-non-euxinic waters has been added. Note that the term ferruginous has been removed throughout the text and replaced with the more general "non-euxinic anoxic" which encompasses suboxic → sub-reducing conditions (after Algeo & Li, 2020) as well as ferruginous conditions.

[Reviewer #2] L133: add (Cd) after "cadmium".

Corrected.

[Reviewer #2] L138-140: this sentence could be misunderstood, I would change it to "Based on its local" and take out the "its" before "deposition", just to make it clear that you are talking about the Gordondale member's regime & deposition because when I first read it, I thought you were trying to say the regime's deposition which didn't make sense.

Corrected.

[Reviewer #2] L229-230: I would change "Lower Pre-N-CIE and Upper..." to "Lower Pre-N-CIE and in the Upper..." because I can see this being misunderstood as the offset between these intervals rather than the offset between the anoxic and euxinic areal extents. I might also add the values of A_{anoxic} and $A_{euxinic}$ in parentheses after "Lower Pre-N-CIE" and "Upper N-CIE".

This section was reworded, and this sentence removed. The comparison between anoxic and euxinic areal extents in each interval (Lower Pre-N-CIE and Upper N-CIE) are now made separately from each other in:

[L325-L328] Lower Pre-N-CIE

[L339-L342] Upper N-CIE

[Reviewer #2] L284: I would add "Vienna" in front of "Pee Dee Belemnite".

[L466] Added.

Reviewer 3

[Reviewer #3] Line 31: marine or oceanic productivity

[L35] Inserted "marine", i.e., "...increased organic marine productivity..."

[Reviewer #3] Lines 121-128: These named intervals are very confusing and hard to keep track of in a paper that already has many acronyms to keep track of. I am unsure of the purpose of even naming these intervals within the Gordondale member. I think the authors should reconsider the purpose of these named intervals as these seem to only be applicable to this drill core with these specific set of TM datasets. Instead, the authors could just refer their TM trends relative to strata or just have a simplified approach with two

label of pre-NCIE and NCIE. Lines 205-207: which stratigraphic intervals are you referring to here? Hard to follow this.

We do agree that the named intervals, though descriptive, are difficult to keep track of. However, we have labeled these intervals in this manner to enable assessment via the models with respect to the T-OAE (which we define as the interval covered by the N-CIE). Additionally, the variations in local redox conditions from euxinic → non-euxinic → euxinic up core are not aligned with the transition into the N-CIE. Thus, we have divided the section into these four intervals based both on coincidence with the N-CIE (i.e., *Pre-N-CIE* and *N-CIE*) and local redox in the lowermost or uppermost portions of these two intervals. Local redox plays an important role in how the mass balance models can be applied, as the Mo model must make use of samples deposited from euxinic bottom waters (Lower *Pre-N-CIE* and Upper *N-CIE* intervals). These subdivisions also help us to evaluate the global ocean redox variations at finer temporal resolution using Re. We now also note that these are informal subdivisions.

[Reviewer #3] Figure 2: Why are these lower and upper boundaries of preNCIE and NCIE chosen? They seem to be right at the geochemical breaks in the TM dataset the authors have presented. Are these official stratigraphic designations that others use? If not, then need a sentence or two defining these L & U intervals. Not sure I would call the end of the NCIE at 1575m as this is only based up on one data point where C-13 varies by less than 1‰ between the underlying data point. This could be the end of the NCIE but could also be an outlier of these bulk ¹³C_{org} data. How are suboxic and anoxic defined and differentiated?

Added several additional points above 1575 m to show more definitive end of N-CIE and to show shift to oxic/dysoxic conditions (collapse in trace metal enrichments—not likely to be fully oxidizing as several weight % C_{org} is still preserved).

The intervals were selected, as assumed by the reviewer, based on changes in geochemical signatures. These are not official stratigraphic designations as they were selected to apply the global mass balance models with respect to the N-CIE. This is discussed in the section “Local paleoenvironmental setting of the T-OAE in British Columbia”, where we initially separate the intervals as “*Pre N-CIE*” (i.e., all samples before the start of the N-CIE) and “*N-CIE*” (i.e., during the N-CIE interval). Upon characterizing the local redox environment, we note that the intervals before (*Pre N-CIE*) and during (*N-CIE*) the N-CIE were likely deposited under conditions that fluctuated from suboxidizing to euxinic. We can observe in Figure 2 that large swings in Mo/Al and V/Al occur at ~1585 m (within the *Pre N-CIE* interval) and ~1577 m (within the *N-CIE* interval). Thus, we break the two main intervals into *Lower* and *Upper* subdivisions to provided increased resolution in the global mass balance reconstructions. We state that these are informal subdivisions.

[Reviewer #3] Figure 3: Would recommend rearranging these plots so they are in stratigraphic (i.e., time order) with "pre-NCIE" model results first then "NCIE" results second. This will make things flow more logically with text discussion. Would recommend adding a note here in the caption as to why no "Upper pre-NCIE" and "Lower NCIE" model runs. As you have to refer back to figure 2 to figure this out...i.e., No enrichments in [Mo] through these intervals.

Altered the figure to instead show authigenic Re and Mo enrichments and resulting global anoxic and euxinic seafloor areas from the models in a stratigraphic profile. This should help readers to identify and follow the temporal changes in global redox that are discussed in the text more easily.

MINOR REVISIONS (SUPPLEMENTAL INFO)

[Reviewer #1] Line 137: The Mo/U co-variation is convincing, but the Cd/Mo isn't. This is because the entire Cd/Mo dataset cluster around the cut-off for productivity versus preservation-driven settings and doesn't

clearly indicate upwelling as argued. Do the author have Mn and Co data to produce a full Cd/Mo/Co/Mn cross-plot, which is more informative?

Modified the figure to show the full Cd/Mo vs. Mn*Co plot, which demonstrates that the studied section was most likely deposited in a regime with elevated productivity compared to preservation, mainly in an open-ocean setting (i.e., top left quadrant defined by Sweere et al. 2016).

[Reviewer #2] L75: Is there a reason you use Mo/U ratio in the SI but the U/Mo ratio in the main paper?

In the main text, the U/Mo ratio cited is from Zhou et al. (2012). In the SI, the Mo/U ratio is based on the covariation diagram of Algeo & Tribovillard (2009). We have modified the main text ratio to "Mo/U" to remain consistent with the covariation figure presented in the SI. We re-evaluated thresholds for this and other proxies in the overhauled local redox section.

[Reviewer #2] When you say "seawater ratio" on Lines 75, 76, 79, and 83, do you mean the modern seawater Mo/U ratio, and if so, do you define this anywhere (not the value, just the meaning) because I found it a bit confusing.

Added "modern" and replaced "ratio" with "Mo/U_(aq)" as this better demonstrates our meaning as the dissolved Mo vs U ratio in modern seawater.

[Reviewer #2] L185: add "in"-> "...increase in continental weathering rate..."

Added.

[Reviewer #2] L372: take out "to cause" -> "...may have been a higher..."

Removed.

Reviewer #1 (Remarks to the Author):

I have reviewed the revised version of this paper and commend the authors for making a number of considered changes to their work. I am satisfied that the major comments have been dealt with and recommend publication.

Reviewer #2 (Remarks to the Author):

Overall, I am very impressed by this work especially the careful revisions made after the first round of review. I recommend it is accepted for publication after minor edits.

Line numbers are based on version without tracked changes

L9. Spelling of "hypothesized"

L82. Seems like this sentence "Thallium has an even shorter residence time than Re but to the point where sedimentary Tl isotope data is vulnerable to being influenced by basin restriction" could be worded better. Maybe: "Thallium has an even shorter residence time than Re making sedimentary Tl isotope data vulnerable to influence by basin restriction" or something like that if I understand the meaning correctly?

L142. Not sure if that "Click or tap here to enter text" will show up on the final version, but just wanted to point it out to make sure it would be removed.

L217. Comma after "study section" before "which..."

L265. Sentence starting with "When ORM..." seems repetitive with statement at L129-133 as well as

L255-259. I feel it could go straight from "...ocean circulation." To "The dissolved..."

L306. Replace 'is' at the end of the line with 'are' for the Re concentrations.

L349-364. I don't have a suggestion about these lines, I just wanted to say that this paragraph really cleared a lot up for me regarding the rest of the paper and the findings, so well done on this part, it provides a lot of perspective.

L390. "suggesting" should be "suggests"

L437. What does past "climate events" mean? Suggest changing that to either 'climate change events' or 'mass extinction events' to be more specific.

L699. I assume this will get caught in the publication process, but you forgot the year for this ref: 2015 I believe.

Reviewer #3 (Remarks to the Author):

This is a re-review of a revised manuscript by Kunert and Kendall. The manuscript was a very interesting read in its initial version, with the Re and Mo mass balance model novel and insightful to this interval in Earth history. I have now read the revised version, the response to reviewer comments (both general comments and line notes), and the revised supplementary information.

The authors have very carefully considered all the reviewer comments, and have clearly addressed the issues in both the response to reviewers and in adding substantial material to main text of the manuscript as well as the supplementary information. I feel the better integration with previous global redox proxy data and previous work on modern-ocean reducing continental margin settings

to be well done. I still find the named intervals the authors are using a bit clunky but I accept the authors rationale for using them.

I would deem the manuscript now suitable for publication, pending any desired editorial revisions.

Dear Reviewers,

Thank you again for your feedback and your positive reviews of the updated manuscript. We have taken your recommendations into account during the final revision. Please see our responses to your comments below.

In our responses, the line numbers are based on the new revised version without track changes.

REVIEWERS' COMMENTS

Reviewer #1 (Remarks to the Author):

I have reviewed the revised version of this paper and commend the authors for making a number of considered changes to their work. I am satisfied that the major comments have been dealt with and recommend publication.

As above, thank you for the positive review of our revised manuscript.

Reviewer #2 (Remarks to the Author):

Overall, I am very impressed by this work especially the careful revisions made after the first round of review. I recommend it is accepted for publication after minor edits.

As above, thank you for the positive review of our revised manuscript.

L9. Spelling of "hypothesized"

Corrected to the Oxford spelling.

L82. Seems like this sentence "Thallium has an even shorter residence time than Re but to the point where sedimentary Tl isotope data is vulnerable to being influenced by basin restriction" could be worded better. Maybe: "Thallium has an even shorter residence time than Re making sedimentary Tl isotope data vulnerable to influence by basin restriction" or something like that if I understand the meaning correctly?

Agreed. Changed to "Thallium has an even shorter residence time than Re making sedimentary Tl isotope data potentially vulnerable to influence by basin restriction."

L142. Not sure if that "Click or tap here to enter text" will show up on the final version, but just wanted to point it out to make sure it would be removed.

Thanks for pointing this out. I used the Mendeley plug-in for referencing and it doesn't seem to get along with the Track Changes function very well. The "click or tap..." box doesn't appear in Word but must be present in the PDF. I believe it should be removed now after deleting the reference with Track Changes turned off.

L217. Comma after "study section" before "which..."

Corrected.

L265. Sentence starting with "When ORM..." seems repetitive with statement at L129-133 as well as L255-259. I feel it could go straight from "...ocean circulation." To "The dissolved..."

True, redundant especially with L129-133. This sentence was removed.

L306. Replace 'is' at the end of the line with 'are' for the Re concentrations.

Corrected.

L349-364. I don't have a suggestion about these lines, I just wanted to say that this paragraph really cleared a lot up for me regarding the rest of the paper and the findings, so well done on this part, it provides a lot of perspective.

Glad to provide clarification.

L390. "suggesting" should be "suggests"

Corrected.

L437. What does past "climate events" mean? Suggest changing that to either 'climate change events' or 'mass extinction events' to be more specific.

Given that we cite Ceballos et al. 2015, who discuss the rate of extinction and argue for a new mass extinction event, we favour the use of the term "mass extinction events".

L699. I assume this will get caught in the publication process, but you forgot the year for this ref: 2015 I believe.

Citation #79. This was updated, and it was also missing the journal name and volume! Also added. Updated with Mendeley, and the formatting changed slightly (an indent was removed from the list), so this is now on line 671 in no-markup version.

Reviewer #3 (Remarks to the Author):

This is a re-review of a revised manuscript by Kunert and Kendall. The manuscript was a very interesting read in its initial version, with the Re and Mo mass balance model novel and insightful to this interval in Earth history. I have now read the revised version, the response to reviewer comments (both general comments and line notes), and the revised supplementary information.

The authors have very carefully considered all the reviewer comments, and have clearly addressed the issues in both the response to reviewers and in adding substantial material to main text of the manuscript as well as the supplementary information. I feel the better integration with previous global redox proxy data and previous work on modern-ocean reducing continental margin settings to be well done. I still find the named intervals the authors are using a bit clunky but I accept the authors rationale for using them.

I would deem the manuscript now suitable for publication, pending any desired editorial revisions.

Thank you for your overall positive review. We appreciate your point on the naming scheme for our intervals, which you brought up in your first review of the manuscript. We agree that the names are clunky, and after careful consideration have decided to change the names to "Model Intervals" (MI) 1-4 from oldest/deepest (MI-1) to youngest/shallowest (MI-4).